# Design Particularities of Quadrature Chaos Shift Keying Communication System with Enhanced Noise Immunity for IoT Applications

**DOI:** 10.3390/e27030296

**Published:** 2025-03-12

**Authors:** Darja Cirjulina, Ruslans Babajans, Deniss Kolosovs

**Affiliations:** Institute of Photonics, Electronics and Telecommunications, Riga Technical University, 6A Kipsalas Street, LV-1048 Riga, Latvia; ruslans.babajans@rtu.lv (R.B.); deniss.kolosovs@rtu.lv (D.K.)

**Keywords:** nonlinear systems, chaos shift keying, chaos oscillator, chaotic synchronization, communication system, signal processing, internet of things

## Abstract

This article is devoted to the investigation of synchronization noise immunity in quadrature chaos shift keying (QCSK) communication systems and its profound impact on system performance. The study focuses on Colpitts and Vilnius chaos oscillators in different synchronization configurations, and the reliability of the system in the particular configuration is assessed using the bit error rate (BER) estimation. The research considers synchronization imbalances and demonstrates their effect on the accuracy of data detection and overall transmission stability. The article proposes an approach for optimal bit detection in the case of imbalanced synchronization and correlated chaotic signals in data transmission. The study practically shows the importance of the proposed decision-making technique, revealing that certain adjustments can significantly enhance system noise resilience.

## 1. Introduction

Expanding networks of Internet of Things (IoT) devices transform traditional environments into smart ecosystems in today’s increasingly interconnected world. These systems, incorporating countless sensors and devices, continuously collect and transmit sensitive data across networks, enabling advanced functionalities in healthcare, smart cities, and industrial automation. However, the massive scale of IoT device deployments introduces substantial security challenges. According to the Ericsson Mobility Report [1], the number of IoT-connected devices is expected to rise sharply, driven by advancements in 5G/6G technologies and the increased adoption of new applications. This surge of devices highlights the need for robust security measures to protect sensitive data from unauthorized access. Ensuring secure data transfer in IoT systems is critical for maintaining security and enabling the full potential of these transformative technologies.

In IoT networks, data transfer requires secure methods to protect sensitive information from breaches. Typical data transfer security approaches include encryption, fog computing, or software-defined networking (SDN), each applied to specific layers of the IoT system architecture [2].

Encryption is one of the most widely used methods, and it is applied primarily at the network and application layers. It ensures that data transmitted across IoT networks, if intercepted, are not readable. Symmetric and asymmetric encryption algorithms like Advanced Encryption Standard and RSA (Rivest–Shamir–Adleman) are used. However, the main drawback is the required computational overhead, which can be problematic for resource-constrained low-power IoT devices. In some cases, lightweight encryption protocols are used to balance the trade-off between security and efficiency [3,4,5].

Fog computing enhances IoT security by enabling real-time processing closer to the network edge, reducing latency, and allowing encryption, data filtering, and threat detection before forwarding data to centralized servers. This approach improves data transfer security, particularly in environments requiring rapid decision making; however, it also introduces complexities in managing distributed resources and ensuring security across multiple edge devices [3,6,7]. In addition, SDN provides a network architecture that allows centralized control of data traffic in IoT networks. By enabling network administrators to enforce security policies and dynamically adjust traffic flows in real time, SDN enhances network security and provides better defense against cyber-attacks. However, this centralized control also introduces a single point of failure, meaning that if the central controller is compromised, the entire network’s security could be at risk [8].

While traditional IoT security relies heavily on encryption and network-layer protections [9], physical layer security (PLS) has emerged as a promising alternative for securing resource-constrained devices by leveraging the inherent properties of the communication medium and device-specific characteristics [10,11,12]. By exploiting the randomness and physical properties of the transmission channel, such as channel state information, PLS enhances security without relying on computationally intensive cryptographic operations, making it particularly suitable for IoT devices with limited processing power. Techniques such as artificial noise injection and radio frequency fingerprinting further strengthen security by disrupting eavesdroppers and authenticating devices based on unique hardware characteristics [13,14].

Among the promising developments in PLS is the integration of chaos theory, which enhances security by leveraging the inherent unpredictability of chaotic signals. Chaos-based communication techniques offer a compelling solution for secure data transmission, particularly in IoT sensor networks, where devices have limited computational and power resources. Unlike conventional encryption-based security methods, chaos-based communication systems rely on simple analog chaos oscillators, making them well-suited for energy-efficient and lightweight implementations. These systems enable secure transmission from low-power sensors to larger network nodes without the need for complex operations, addressing key challenges in resource-constrained IoT environments [10]. Chaotic systems such as the Lorenz and Chua circuits have been effectively utilized in chaos-based communication systems to secure data transmissions [15,16,17]. These systems generate chaotic waveforms characterized by high sensitivity to initial conditions and noise-like waveforms, making them ideal for encoding sensitive information. In such implementations, the transmitter and receiver must achieve precise synchronization to decode the transmitted data, providing an additional layer of security. Any disruption in synchronization renders the intercepted signal unreadable, significantly complicating unauthorized access [18,19].

Chaos-based modulation techniques, such as differential chaos shift keying (DCSK), further enhance PLS by resisting eavesdropping. These methods utilize chaotic signals’ wideband and nonperiodic nature, making them particularly suited for IoT environments where low-power, high-security solutions are required. Unlike traditional key-based approaches, chaos-based systems inherently ensure difficult-to-intercept without needing pre-shared keys, reducing overhead and simplifying implementation [14,19].

Recent research has further expanded the applicability of chaos-based communication systems, particularly in multi-carrier and MIMO configurations, to enhance security and spectral efficiency. Multi-carrier implementations of chaos shift keying (CSK) and its derivatives improve signal robustness while maintaining the inherent unpredictability of chaotic waveforms, making them resistant to eavesdropping and interference [20]. Additionally, synchronization stability remains a critical aspect, as efficient drive-response synchronization enables accurate data recovery even in noisy environments. Studies on breaking time correlation among state variables have demonstrated enhanced security by minimizing the predictability of transmitted signals, further reinforcing chaos-based methods as viable solutions for secure IoT communication [21].

Researchers are developing advanced chaotic models to overcome limitations in existing systems, such as lower complexity and improved sensitivity and resilience to noise. Innovations like the Exponential Chaotic Model have introduced robust chaos, ensuring reliability even in noisy channels and offering new possibilities for secure communication [22,23]. Beyond communication, chaos has found applications in multimedia security and random number generation. Advanced image encryption algorithms based on dynamic chaotic systems provide robustness against statistical and differential attacks and ensure data integrity in IoT environments [24,25,26,27].

While the integration of chaos into PLS represents a significant leap forward, challenges remain. Active research areas include synchronization under varying channel conditions, efficient implementation in resource-limited devices, and the potential vulnerabilities of chaotic systems. Despite these constraints, chaos-based PLS offers a compelling pathway for IoT security, blending innovation with practicality to address the unique challenges of modern networks [28,29].

Synchronization is a crucial part of chaos-based communication systems, which allows them to safely send and receive data in IoT and wireless sensor networks. At its core, chaotic synchronization ensures that the state variables of a receiver’s chaotic oscillator replicate those of the transmitter’s oscillator, enabling precise signal decoding. Techniques like the Pecora–Carroll method [30] and adaptive control [31,32] are commonly used to achieve synchronization by coupling specific state variables between drive and response oscillators.

Research on the topic has consistently emphasized the critical role of synchronization stability in the performance of chaos-based systems. For instance, studies on systems such as the Vilnius chaos oscillator highlight the importance of robust synchronization methods to withstand noise, parameter deviations, and environmental variability [33]. This is particularly crucial in applications of chaos-based communication, where synchronization directly impacts the system’s ability to securely and accurately transmit data [34,35].

Our previous studies have examined key synchronization parameters, including synchronization time and noise immunity [35]. These investigations provided valuable insights but were often limited to single configurations. Additionally, research into analog-discrete synchronization has explored different configurations, highlighting the potential for hybrid systems to achieve robust synchronization under specific conditions [36,37].

In this work, we extend this research by testing chaos oscillator synchronization in noisy environments to evaluate its resilience and behavior in various configurations. The collected experimental data were processed and analyzed in MATLAB R2023a to assess synchronization performance under different noise conditions. Based on these findings, we propose an approach for quadrature chaos shift keying (QCSK) communication system construction, where the system’s core is a chaos oscillator and secure communication is ensured through chaotic synchronization between the drive and response oscillators. QCSK utilizes chaos shift keying (CSK) to encode the data stream, with quadrature amplitude modulation applied to enhance spectral efficiency. This approach offers improved resistance to noise and eavesdropping compared to conventional CSK schemes. This study aims to evaluate the performance of the QCSK system and determine its design particularities by assessing synchronization noise immunity across different configurations and analyzing the bit error rate (BER) in the QCSK system. The research addresses the limitations of prior studies by providing a comprehensive evaluation of synchronization methods, bridging the gap between theoretical models and practical applications, and advancing the field of chaos-based secure communications.

This article is organized into five sections. Section 2 introduces the Colpitts and Vilnius chaos oscillators and provides an overview of chaotic synchronization. Section 3 investigates the noise immunity of synchronization between chaotic oscillators across different configurations. Section 4 examines the QCSK communication system, evaluating its noise immunity. Finally, Section 5 concludes the study.

## 2. Chaos Oscillators

Chaotic signals demonstrate deterministic aperiodic behavior, sensitivity to initial conditions, and a wide spectrum, making them highly appealing for communication and improved security applications. Chaotic signals emerge when nonlinear oscillators are set to work in certain conditions, with close-tuned component nominals forcing the system into the chaotic region. This study is dedicated to the Colpitts and Vilnius chaos oscillators, two of the most popular devices in chaos research [38,39,40,41]. The other crucial issue of applying chaotic signals in real-world applications is synchronization, which is the process of achieving the coincidence of chaotic signals at the transmitter and receiver end for reliable and secure data transfer [42,43].

### 2.1. Colpitts Chaos Oscillator

The chaos oscillator under investigation is a nonlinear dynamical system that produces chaotic signals distinguished by their sensitivity to initial conditions and noise-like waveforms. These oscillators are critical in secure communication systems because their chaotic nature is used to ensure PLS. Figure 1 shows the oscillator’s circuit diagram, which includes components such as resistors, capacitors, inductors, and a nonlinear element. The system shows chaotic dynamics under certain parameter values [44].

The oscillator’s chaotic behavior can be mathematically characterized by a set of differential equations based on its circuit design. Let $v_{CE}$, $v_{BE}$, and $i_{L1}$ represent the voltages across the capacitors $C_{1}$ and $C_{2}$ and the current through the inductor $L_{1}$. The state equations are stated as follows:(1)$$\left\{\begin{array}{c} C_{1}\frac{dv_{CE}}{dt}=i_{L}-i_{C}\\ C_{2}\frac{dv_{BE}}{dt}=\frac{V_{1}+v_{BE}}{R_{1}}-i_{L}-i_{B}\\ L_{1}\frac{di_{L1}}{dt}=V_{2}-v_{CE}+v_{BE}-i_{L}\cdot R_{L} \end{array}\right.,$$
where $v_{CE}$ corresponds to the collector-emitter voltage, $v_{BE}$ the base-emitter voltage, $V_{1}$ and $V_{2}$ are the source voltages, $i_{L}$ is the inductor current, $i_{C}$ is the collector current, and $i_{B}$ is the base current.

The oscillator’s chaotic behavior depends on its nonlinear element. A piecewise linear function approximates the nonlinear voltage-controlled resistance the transistor provides in the Colpitts oscillator. This function is described by:(2)$$i_{B}=\left\{\begin{array}{c} 0,\;\mathrm{if}\;v_{BE}\leq V_{TH}\\ \frac{v_{BE}-V_{TH}}{R_{ON}},\;\mathrm{if}\;v_{BE}>V_{TH} \end{array}\right.,$$(3)$$i_{C}=\beta_{F}\cdot i_{B},$$
where $V_{TH}$ denotes the threshold voltage, $R_{ON}$ represents the small-signal on-resistance of the base-emitter, and $\beta_{F}$ indicates the transistor’s forward current gain.

### 2.2. Vilnius Chaos Oscillator

The Vilnius chaos oscillator is a simple and robust nonlinear system for generating chaotic signals. It is widely recognized for its simplicity and effectiveness, making it suitable for research and educational applications in nonlinear dynamics. The circuit, shown in Figure 2, consists of a noninverting operational amplifier, an LCR resonance circuit in a positive feedback loop, an additional capacitor, and a diode functioning as the nonlinear element. This configuration ensures chaotic dynamics with minimal components while facilitating experimental implementation and analysis.

The behavior of the Vilnius chaotic oscillator is characterized by three state variables: $v_{C1}$, the voltage across the capacitor $C_{1}$; $i_{L}$, the current through inductor $L_{1}$; and $v_{C2}$, the voltage across the supplementary capacitor $C_{2}$. The system is defined by an array of differential equations derived using Kirchhoff’s circuit laws:(4)$$\left\{\begin{array}{c} C_{1}\frac{dv_{C1}}{dt}=i_{L}\\ C_{2}\frac{dv_{C2}}{dt}=i_{0}+i_{L}-i_{D}\\ L_{1}\frac{di_{L}}{dt}=\left(k-1\right)\cdot R_{1}\cdot i_{L}-v_{C_{1}}-v_{C_{2}} \end{array}\right.,$$
where $i_{D}$ is the current of the diode $D_{1}$, $i_{0}$ is the current of the resistor $R_{0}$, and *k* is the gain of the operational amplifier.

The oscillator’s nonlinear behavior is defined by the diode’s current-voltage characteristic, represented by the subsequent equation:(5)$$i_{D}=i_{S}\cdot \left(e^{\frac{v_{D}}{v_{T}}}-1\right),$$
where $v_{D}$ is the voltage across the diode and due to parallel connection $v_{D}=v_{C2}$, $i_{S}$ is the saturation current of the diode, and $v_{T}$ is the thermal voltage.

This nonlinear characteristic, combined with the circuit design, facilitates the onset of chaotic oscillations. The system exhibits a universal period-doubling route to chaos, as demonstrated through bifurcation diagrams and positive Lyapunov exponents, confirming its chaotic nature under suitable parameter conditions [41].

### 2.3. Chaotic Synchronization

Chaotic synchronization is fundamental for utilizing chaotic oscillators in secure communication systems [47]. It enables the secure transmission of information by ensuring that only synchronized systems can accurately extract data from received signals. Among the various synchronization methods, the substitution method, also known as Pecora–Carroll synchronization [48], is particularly noteworthy for its simplicity and effectiveness in both analog and discrete chaotic systems.

The substitution method replaces one of the state variables in the response chaotic oscillator with its corresponding variable from the drive oscillator. In this approach, the response oscillator is driven by a signal from the drive circuit, aiming to align their chaotic trajectories. Pearson’s correlation coefficient is often employed to evaluate the quality of synchronization. This metric quantifies the similarity between the signals from the drive and response oscillators. The coefficient, denoted as β, is calculated using the formula:(6)$$\beta =\frac{\sum_{i=1}^{n}\left(x_{i}-\bar{x}\right)\left(y_{i}-\bar{y}\right)}{\sqrt{\sum_{i=1}^{n}{\left(x_{i}-\bar{x}\right)}^{2}\sum_{i=1}^{n}{\left(y_{i}-\bar{y}\right)}^{2}}},$$
where *x* is the signal from the drive oscillator and *y* is the signal from the response circuit. If the signals are identical, β equals 1; if they are inverted, β is −1; and if β is 0, the signals are uncorrelated, indicating no similarity.

The same node is selected for synchronization in both the drive and response chaos oscillators to implement the substitution method. An operational amplifier configured as a voltage repeater (buffer circuit) is commonly used for analog chaos oscillator synchronization [49,50,51]. This configuration determines the direction of synchronization and allows the signal in the response circuit to be substituted with the corresponding signal from the drive oscillator. Since chaotic oscillators typically have multiple nodes, synchronization can be achieved in various configurations by selecting different nodes for substitution.

An example illustrating unsynchronized and synchronized signals is presented in Figure 3. In Figure 3a, the XY plot shows the unsynchronized signals from the same node of identical chaos oscillators. The lack of order is noticeable, as the data points are scattered randomly. In contrast, Figure 3b presents an XY plot of a synchronized drive-response system. The x-axis represents the signal from the drive chaos oscillator, while the y-axis shows the signal from the same node in the response oscillator. The data points align along a diagonal line, demonstrating that the voltage in the response oscillator corresponds precisely to the voltage in the drive oscillator. Such a result is indicative of a successfully synchronized system.

In the Colpitts chaos oscillator (Figure 1), synchronization can be established using two configurations, i.e., $v_{C1}$ or $v_{C2}$, as synchronization signals [52]. For instance, if $v_{C1}$ is chosen for synchronization, the drive oscillator generates all three state variables ($v_{C1}$, $v_{C2}$, and $i_{L1}$), while the response oscillator generates $v_{C2}$ and $i_{L1}$, with $v_{C1}$ being substituted by the corresponding signal from the drive oscillator. In the Vilnius chaos oscillator (Figure 2), synchronization can be achieved using $v_{C1}$, $v_{C2}$, or $v_{R1}$ (representing the current $i_{L}$) as a synchronization signal. These multiple synchronization points offer flexibility for future applications.

As synchronization in chaotic oscillators forms the basis for their integration into communication systems, it is essential to assess the noise resilience of synchronization across various configurations. Understanding how synchronization performance varies in noisy environments is critical for ensuring reliable communication. These experiments and their results will be discussed in the following section.

## 3. Chaos Oscillator Synchronization Noise Immunity

This study investigates the noise immunity of chaotic synchronization for both the Colpitts and Vilnius chaos oscillators. Different noise levels and circuit configurations are considered for synchronization performance assessment. Knowing how synchronization works in noisy environments is important for utilizing it in secure communication and signal processing.

### 3.1. Study Methodology

The study methodology follows the block diagram presented in Figure 4. In this figure, *X*, *Y*, and *Z* represent the state variables in the drive chaos oscillator, while $X^{\prime }$, $Y^{\prime }$, and $Z^{\prime }$ denote the corresponding state variables in the response chaos oscillator.

In this setup, *Z* is selected as the synchronization signal, replacing $Z^{\prime }$ in the response oscillator. The *X* and *Y* signals generated by the drive oscillator will be employed in the final data transmission system for chaos shift keying (CSK). Once the drive oscillator generates the signals, they are transmitted through an additive white Gaussian noise (AWGN) channel, where the noise level is controlled between −20 dB and 30 dB in terms of the signal-to-noise ratio (SNR).

At the receiver side, the signals pass through a low-pass filter (LPF), which is specifically tuned to preserve the main spectral components of the chaotic signal. The LPF bandwidth is determined by comparing the original chaotic signal (before noise addition and filtering) with the filtered version (without noise). The bandwidth is selected when the mean square error (MSE) reaches −25 dB, ensuring the signal maintains its chaotic properties. After filtering, the synchronization signal substitutes the variable in the drive-response chaotic system to ensure the oscillators are in the same state. Once synchronization is achieved, the Pearson correlation coefficient is computed for the corresponding state variables: *X* and $X^{\prime }$, *Y* and $Y^{\prime }$, and $X^{\prime }$ and $Y^{\prime }$. This analysis provides insights into the effect of noise on chaotic synchronization in the transmission channel.

The study is conducted for both the Colpitts and Vilnius chaos oscillators. Since each chaotic oscillator has three state variable signals, all of them are tested for synchronization capability. Table 1 summarizes the different configurations tested. For the Colpitts chaos oscillator (Figure 1), synchronization is tested using $v_{C1}$, $v_{C2}$, and $v_{RL}$, while for the Vilnius chaos oscillator (Figure 2), synchronization is evaluated using $v_{C1}$, $v_{C2}$, and $v_{R1}$. The remaining signals are analyzed to assess synchronization quality across different configurations.

By systematically testing noise immunity across multiple configurations, this methodology provides knowledge on the robustness of chaotic synchronization.

### 3.2. Result Analyses

This study analyzes the performance of Colpitts and Vilnius chaos oscillators in different drive-response system configurations within a noisy channel. The goal is to evaluate synchronization performance across different configurations and identify the most robust configuration.

Figure 5 presents the study results for the Colpitts chaos oscillator. The *x*-axis represents the SNR levels but is flipped to encourage a more convenient interpretation. The *y*-axis represents the correlation coefficient, which quantifies the synchronization quality. The figure contains six curves, each pair corresponding to a specific configuration, as summarized in Table 1. Each pair represents the correlation between *X* and $X^{\prime }$ and *Y* and $Y^{\prime }$, illustrating how synchronization is maintained under different noise conditions.

Examining Figure 5, it can be observed that $X^{\prime }$ and $Y^{\prime }$ signals synchronize differently for each SNR level. When $v_{RL}$ is used as the synchronization signal, the system fails to achieve synchronization in the drive-response configuration. When $v_{C1}$ is employed, $X^{\prime }$ and $Y^{\prime }$ (corresponding to $v_{C2}$ and $v_{RL}$) synchronize at different levels for the same SNR values. A similar trend is observed when $v_{C2}$ is used as the synchronization signal; however, $X^{\prime }$ and $Y^{\prime }$ synchronize at different levels and show smaller correlation coefficients in this configuration.

Figure 6 illustrates the synchronization noise immunity across all three configurations for the Vilnius chaos oscillator. Like Figure 5, it consists of six curves, where each pair represents the correlation between *X* and $X^{\prime }$ and *Y* and $Y^{\prime }$, corresponding to a specific configuration described in Table 1.

When $v_{C1}$ is used for synchronization, the response oscillator signals $v_{C2}$ and $v_{R1}$ do not synchronize at the same level for a given SNR value. In contrast, when $v_{C2}$ is the synchronization signal, $v_{C1}$ and $v_{R1}$ synchronize at the same level, as the two curves overlap entirely. Additionally, this configuration achieves synchronization at lower SNR values than others. When $v_{R1}$ is selected as the synchronization signal, an imbalance in synchronization levels is observed, similar to the case of $v_{C1}$.

Comparing the results for Colpitts and Vilnius oscillators, it is evident that for the Vilnius chaos oscillator, all three configurations achieve synchronization once the noise level is sufficiently low. In contrast, the Colpitts oscillator demonstrates reliable synchronization only in one configuration when $v_{C1}$ is used as the synchronization signal. The synchronization with the considerably lower correlation level occurs in the $v_{C2}$ configuration. Additionally, one configuration exists for the Colpitts oscillator in which synchronization is never achieved.

The results suggest that the Colpitts oscillator exhibits a more restrictive synchronization behavior due to its circuit topology, nonlinear characteristics, and strong variable coupling, which limit the effectiveness of synchronization in some configurations. In contrast, the Vilnius chaos oscillator provides more flexible synchronization possibilities, allowing multiple high-correlation configurations.

Another important observation is the presence of synchronization imbalances across different configurations for oscillators at the same SNR level. This imbalance can negatively impact data transmission since one signal may be more likely to be detected correctly, leading to an increased error rate for specific bits. For example, an imbalance in synchronization levels may result in a scenario where more ‘0’ s are incorrectly detected compared to ‘1’ s. It is a point to improve the chaos-based communication system so that each bit is detected with equal probability to ensure a balanced transmission process.

The next part of this study examines the cross-correlation of response chaos oscillator signals as a function of the synchronization channel SNR. This analysis extends the findings from the previous part by evaluating how the relationship between $X^{\prime }$ and $Y^{\prime }$ signals evolves across different configurations. The cross-correlation value between these signals is crucial for maintaining balance in a data transmission system, as imbalances can lead to an increased number of errors.

Figure 7 presents the study results for the Colpitts chaos oscillator, illustrating the cross-correlation coefficients under different synchronization configurations.

The results indicate that when $v_{RL}$ is used as the synchronization signal, the cross-correlation coefficient remains close to one, meaning the response chaos oscillator signals exhibit similar dynamics. However, in this configuration, the Colpitts chaos oscillator correlation coefficient of data-carrying signals was of nearly zero level in the drive-response system, leading the response oscillator to enter a periodic mode instead of remaining chaotic. For the other two synchronization signals, $v_{C1}$ and $v_{C2}$, the cross-correlation between response oscillator signals decreases as the noise level decreases. Additionally, it can be observed that when $v_{C1}$ is used as the synchronization signal, the absolute value of the cross-correlation coefficient is lower compared to the $v_{C2}$ configuration.

Figure 8 illustrates the cross-correlation coefficients as a function of the SNR in the Vilnius chaos oscillator for different synchronization signals.

In Figure 8, the cross-correlation coefficient between $X^{\prime }$ and $Y^{\prime }$ decreases as the noise level decreases in configurations where $v_{C1}$ and $v_{R1}$ are used as synchronization signals. A more significant variation in the cross-correlation coefficient is observed when $v_{R1}$ is used, suggesting that synchronization in this configuration is more affected by noise. Conversely, when $v_{C2}$ is used as the synchronization signal, the cross-correlation coefficient remains nearly constant, indicating a strong and noise-resistant synchronization.

Comparing Figure 7 and Figure 8 with previous results (Figure 5 and Figure 6), it is evident that the Vilnius chaos oscillator demonstrates the most robust synchronization when $v_{C2}$ is employed as the synchronization signal. In this case, the correlation between *X* and $X^{\prime }$ and between *Y* and $Y^{\prime }$ is balanced, while the cross-correlation between $X^{\prime }$ and $Y^{\prime }$ remains stable, regardless of the noise level. However, a high cross-correlation level between data-carrying signals complicates signal discrimination in the receiver in this configuration.

On the other hand, in the Colpitts chaos oscillator, $v_{C1}$ appears to be the most suitable synchronization signal for communication system applications. In this configuration, the absolute value of the cross-correlation coefficient is the lowest of the observed, and its dependence on the noise level is minimal. Additionally, the correlation curves between *X* and $X^{\prime }$ and between *Y* and $Y^{\prime }$ achieve synchronization as rapidly as in the Vilnius chaos oscillator when $v_{C2}$ is used as the synchronization signal.

## 4. Quadrature Chaos Shift Keying Communication System

This section presents the architecture, design particularities, and noise immunity analysis of the QCSK data transmission system across various configurations and different chaos oscillators. The objective is to evaluate how the choice of chaotic oscillator—specifically, the Vilnius and Colpitts chaos oscillators—and the selection of the synchronization signal influence the overall performance of the data transmission system.

### 4.1. System Design Particularities

The QCSK communication system is based on chaos shift keying and chaotic synchronization. The core elements of the transmitter and receiver are chaos oscillators operating in a drive-response configuration. Figure 9 presents the block diagram of the QCSK data transmission system.

At the transmitter, the state variables *X*, *Y*, and *Z* represent the output signals of the drive chaos oscillator. The *X* and *Y* signals are used for CSK modulation, forming the information-carrying signal by switching these chaotic signals in correspondence to data values, while the *Z* signal is used as a synchronization signal. Before CSK and quadrature modulation, a DC offset is canceled in all signals. The synchronization signal and keyed data signals are then used for quadrature modulation and transmitted through the AWGN channel.

At the receiver, the input signals are quadrature demodulated and filtered using the LPF, which is tuned to preserve the main spectral components of the chaotic signal. The LPF bandwidth is selected to ensure the MSE for signals before and after filtering is −25 dB. The synchronization signal is then used to synchronize the response chaos oscillator via the substitution method, as described in the previous section. The received information-carrying signals, along with the response chaos oscillator signals ($X^{\prime }$ and $Y^{\prime }$), are then used for data detection.

Information detection is based on correlation coefficient estimations between the received information-carrying signal and the response chaos oscillator signals $X^{\prime }$ and $Y^{\prime }$. After computing the correlation, a comparator makes a decision based on the correlation coefficient value. Section 3 showed different correlation values for the oscillator’s state variable as well as the nonzero cross-correlation for the signals used for the information-carrying signal forming. Therefore, a nonzero threshold utilization is required for decision making.

For threshold calculation, the likelihood ratio approach is utilized. Given that white Gaussian noise is added to the signal, the likelihood that the received information-carrying signal s(t) was induced by the signal *X* can be expressed in the form: (7)$$\Lambda \left[X|s\left(t\right)\right]=\frac{1}{{\left(\sigma \sqrt{2\pi }\right)}^{K}}\exp\left(-\frac{1}{2}\sum_{k=1}^{K}\frac{{\left(s_{k}-x_{k}\right)}^{2}}{\sigma^{2}}\right),$$
where σ denotes the AWGN standard deviation and $s_{k}$ and $x_{k}$ denote *k*th sample of CSK signal s(t) and chaotic signal *X*, respectively. Then, the logarithm of the likelihood ratio can be expressed as: (8)$$\ln\Lambda \left[s\left(t\right)\right]=\frac{\Lambda [X|s(t)]}{\Lambda [Y|s(t)]}=\sum_{k=1}^{K}\frac{s_{k}}{\sigma^{2}}\left(x_{k}-y_{k}\right)-\frac{1}{2}\sum_{k=1}^{K}\frac{\left(x_{k}^{2}-y_{k}^{2}\right)}{\sigma^{2}}.$$
In this case, the decision-making regions are split by the point where the likelihood ratio changes sign, which is given by: (9)$$\sum_{k=1}^{K}s_{k}\left(x_{k}-y_{k}\right)-\frac{1}{2}\sum_{k=1}^{K}\left(x_{k}^{2}-y_{k}^{2}\right)⩾0.$$
Assuming that the information-carrying signal s(t) is a sum of *X* state variable and noise signal n(t), one can express the mathematical expectation E[] of the first sum as: (10)$$E\left[\left(x_{k}+n_{k}\right)\left(x_{k}-y_{k}\right)\right]=\sum_{k=1}^{K}x_{k}^{2}-\sum_{k=1}^{K}x_{k}y_{k},$$
where $n_{k}$ is the *k*th sample of the noise signal. By substituting the calculated mathematical expectation in Equation (Equation 9), one can conclude that a high cross-correlation between the response oscillator’s state variables $X^{\prime }$ and $Y^{\prime }$ decreases correct detection probability.

Rewriting Equation (Equation 9) gives the following criterion for decision making: (11)$$\sum_{k=1}^{K}s_{k}x_{k}⩾\sum_{k=1}^{K}s_{k}y_{k}+\frac{1}{2}\left[\sum_{k=1}^{K}x_{k}^{2}-\sum_{k=1}^{K}y_{k}^{2}\right].$$
Alternatively, the sums can be replaced by a priori ($\beta_{X}$ and $\beta_{Y}$) and a posteriori ($\beta_{X^{\prime }}$ and $\beta_{Y^{\prime }}$) correlation coefficients, resulting in the criterion in the form $\beta_{X^{\prime }}⩾\beta_{Y^{\prime }}+\frac{1}{2}\left(\beta_{X}-\beta_{Y}\right)$. If inequality is valid, the system detects a ‘1’ bit; otherwise, a ‘0’ bit is received.

### 4.2. Result Analyses

The QCSK communication system was considered in four configurations: three based on the Vilnius chaos oscillator, corresponding to the configurations described in Table 1, and one based on the Colpitts chaos oscillator with $v_{C1}$ as the synchronization signal. As demonstrated in our previous study, the remaining two configurations of the Colpitts chaos oscillator were excluded from further analysis due to their low correlation coefficient values. These four QCSK configurations were tested in an AWGN channel. Table 2 presents an example of falsely detected bit counts and BER values under different SNR conditions, comparing the performance with and without the added detection threshold. These results are provided for the QCSK configuration using $v_{R1}$ as the synchronization signal.

As shown in Table 2, an imbalance emerges at higher SNR values, where correlation coefficient levels become more distinct if the detection threshold is not applied. This imbalance leads to detection errors, as one signal may dominate, increasing the probability of misdetection of ‘1’ or ‘0’ bits. When the threshold is introduced, it compensates for this imbalance, resulting in improved BER performance and a more stable data detection process.

The noise immunity study results for all four configurations are presented in Figure 10, where the *y*-axis represents the BER and the x-axis shows bit energy and the noise power spectral density ratio ($E_{b}/N_{0}$).

The best performance among the tested QCSK configurations was observed in the Vilnius-chaos-oscillator-based system when $v_{C2}$ was used as the synchronization signal. In the chaotic synchronization noise immunity study, this configuration demonstrated the highest correlation values under noisy conditions, ensuring that *X* and *Y* correlations are of the same level. Additionally, the cross-correlation between chaotic signals forming the information-carrying signals was minimal, and it remained relatively unaffected by added noise, further enhancing system’s resilience.

The QCSK system based on the Colpitts chaos oscillator achieved the second-best performance using $v_{C1}$ for synchronization. In the previous study, this configuration displayed similar synchronization characteristics to the $v_{C2}$ synchronized Vilnius chaos oscillator. However, in the Colpitts oscillator, $X^{\prime }$ and $Y^{\prime }$ exhibited different correlation coefficient values, leading to a system imbalance, which required compensation during detection.

The remaining two configurations of the Vilnius chaos oscillator exhibited similar performance trends, aligning with findings from the synchronization noise immunity study. However, the worst BER performance was observed in the Vilnius-chaos-oscillator-based QCSK system when $v_{R1}$ was used as the synchronization signal. This configuration was more noise-sensitive since spectral components of the information-carrying signal occupied a wider band [37].

## 5. Discussion

The results of this study highlight the importance of careful signal selection for QCSK communication systems, particularly when operating under noisy conditions. The results demonstrate that synchronization stability plays a crucial role in signal detection accuracy, directly influencing the BER of the system. The comparison between Vilnius and Colpitts chaos oscillators provides insights into their respective strengths and limitations in chaotic communication systems.

The Vilnius chaos oscillator with $v_{C2}$ as the synchronization signal exhibited the best synchronization robustness, with $X^{\prime }$ and $Y^{\prime }$ signals synchronizing at the same level while maintaining low cross-correlation between information-carrying signals. This configuration also proved to be less sensitive to noise, as its correlation values remained stable even under low SNR conditions. These results suggest that this configuration is optimal for chaos-based communication systems, where maintaining synchronization in noisy environments is critical. However, the high cross-correlation between data-carrying signals in this configuration presents a challenge for signal discrimination at the receiver, which may require further refinement in detection methods.

The Colpitts chaos oscillator with $v_{C1}$ as the synchronization signal demonstrated the second-best performance, closely following the Vilnius $v_{C2}$ configuration in synchronization stability and BER performance. However, in this case, $X^{\prime }$ and $Y^{\prime }$ did not synchronize at equal levels, leading to a system imbalance that required compensation during detection. Despite this, the Colpitts $v_{C1}$ configuration maintained the lowest cross-correlation values, suggesting that it may be preferable in applications where minimizing interference between information-carrying signals is a priority.

In contrast, configurations using $v_{R1}$ as the synchronization signal exhibited the weakest performance, particularly in the Vilnius-oscillator-based QCSK system. This configuration was highly sensitive to noise, leading to a significant degradation in synchronization stability and increased BER. In this case, the wider band of the information-carrying signals made them more susceptible to interference, further reducing transmission reliability. These findings emphasize the importance of careful synchronization signal selection, as it directly impacts system robustness against noise and overall transmission performance.

Furthermore, the results confirm that the imbalance in the synchronization levels negatively affects data detection accuracy, as seen in Table 2. Without a detection threshold, correlation coefficient disparities at higher SNR values led to increased misdetection of ‘1’ and ‘0’ bits, resulting in an uneven bit error distribution. Introducing a threshold-based correction mechanism successfully balanced the system, reducing detection errors and improving BER. These observations underscore the necessity of implementing adaptive compensation techniques in chaos-based communication systems to maintain balanced detection probabilities for reliable data transmission.

Overall, this study demonstrates that synchronization signal selection and system configuration are critical in determining QCSK performance. The Vilnius $v_{C2}$ and Colpitts $v_{C1}$ configurations exhibited the most stable synchronization and lower BER, making them strong candidates for secure and efficient chaos-based communication systems. When designing a chaos-based data transmission system, it is essential to evaluate the noise immunity of chaos oscillator synchronization across all possible configurations. Based on the findings of this study, incorporating a threshold into the detector, when necessary, can optimize system performance, leading to an optimal QCSK configuration.

## 6. Conclusions

This study analyzed the synchronization noise immunity of Vilnius and Colpitts chaos oscillators in different configurations. The results showed that the Vilnius oscillator with $v_{C2}$ as the synchronization signal exhibited the most robust synchronization, maintaining low cross-correlation between information-carrying signals and higher noise resistance. The Colpitts oscillator with $v_{C1}$ also performed well but required compensation during detection due to synchronization imbalance. In contrast, configurations using $v_{R1}$ as the synchronization signal were highly noise-sensitive, leading to weaker synchronization stability and higher BER.

A QCSK communication system was implemented using these synchronization configurations, demonstrating that the choice of synchronization signal directly impacts BER performance. The Vilnius $v_{C2}$ and Colpitts $v_{C1}$ configurations achieved the lowest BER, making them the most suitable for chaos-based communication systems.

A threshold-based correction mechanism was introduced to address synchronization imbalances. Without a threshold, correlation coefficient disparities at higher SNR values caused an imbalance in detected bits. The applied threshold successfully reduced detection errors and improved BER, demonstrating the importance of compensation techniques in ensuring balanced data detection in chaos-based systems. These findings highlight the need to carefully select synchronization configurations and optimize detection strategies to enhance the reliability of QCSK communication systems. 

## Figures and Tables

**Figure 1 entropy-27-00296-f001:**
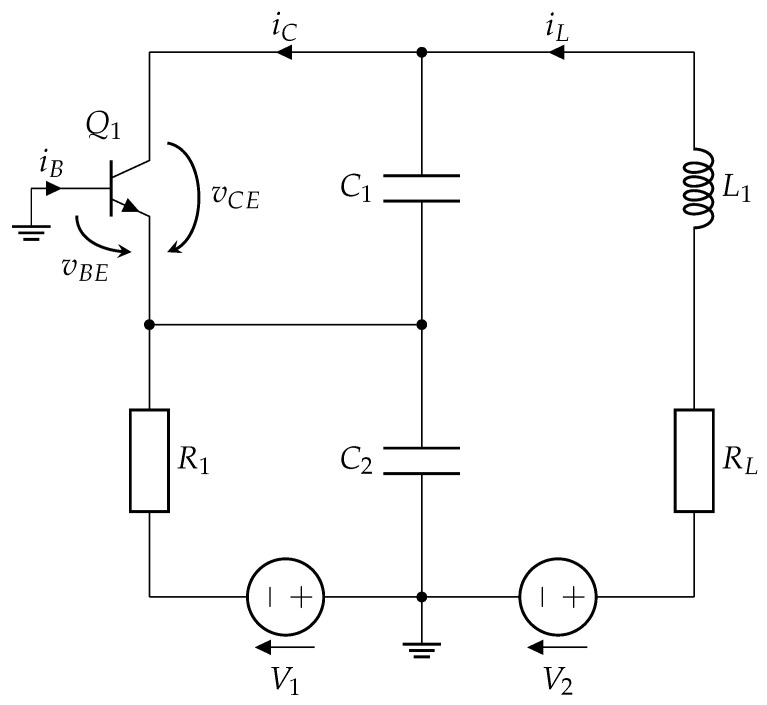
Colpitts chaos oscillator [45] with the following parameters: $R_{L}=35\;\Omega$, $L_{1}=10\;\mu$H, $C_{1}=5.4$ nF, $C_{2}=5.4$ nF, $R_{1}=400$
Ω, $V_{1}=5$ V, $V_{2}=5$ V, with a 2N2222 transistor. These parameters resulted in a fundamental frequency of 968.59 kHz.

**Figure 2 entropy-27-00296-f002:**
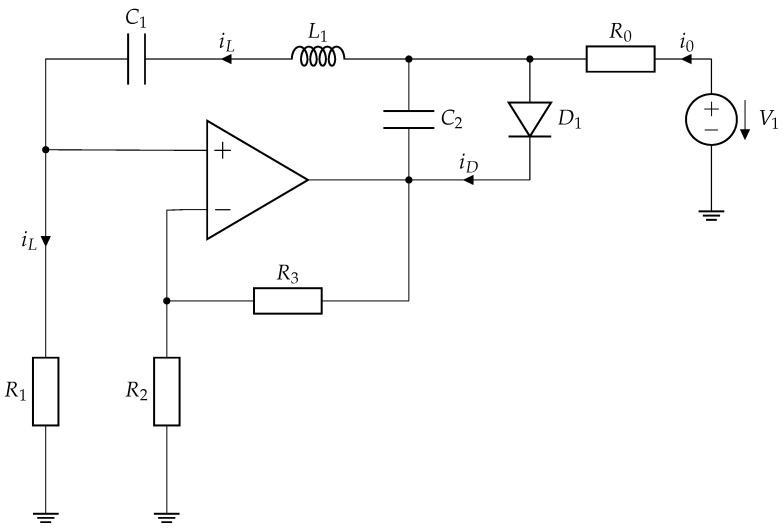
Vilnius chaos oscillator [46] with the following parameters: $C_{1}=1$ nF, $C_{2}=150$ pF, $L_{1}=1$ mH, $R_{1}=1$
kΩ, $R_{2}=1$
kΩ, $R_{3}=600$
Ω, $R_{0}=20$
kΩ, $V_{1}=5$ V, k=1.6. These parameters resulted in a fundamental frequency of 160 kHz.

**Figure 3 entropy-27-00296-f003:**
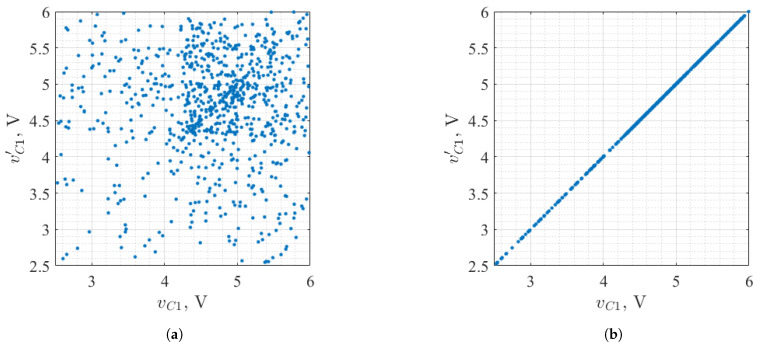
XY plot of drive-response system signals (**a**) without synchronization and (**b**) with chaotic synchronization applied.

**Figure 4 entropy-27-00296-f004:**
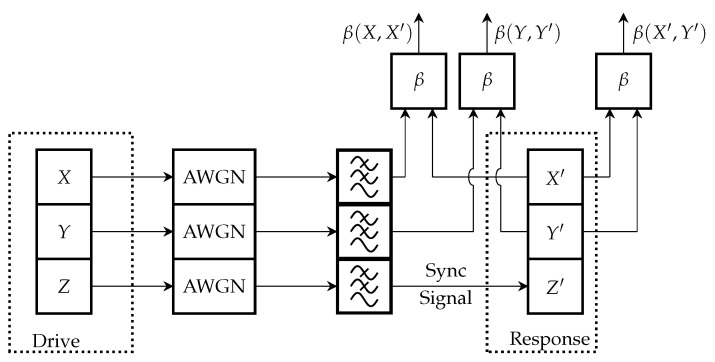
Chaos oscillator synchronization noise immunity study block scheme.

**Figure 5 entropy-27-00296-f005:**
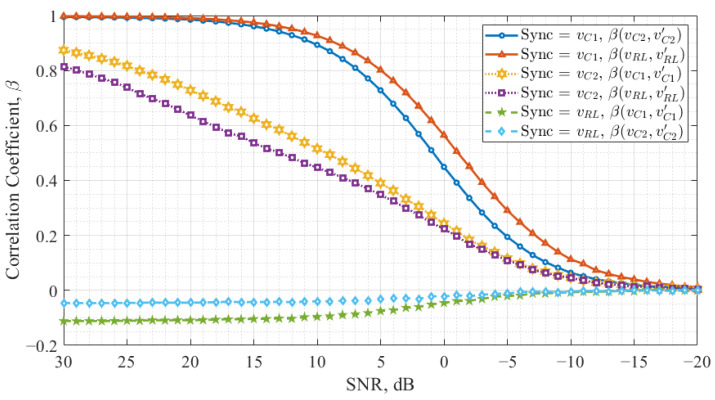
Colpitts chaos oscillator synchronization noise immunity.

**Figure 6 entropy-27-00296-f006:**
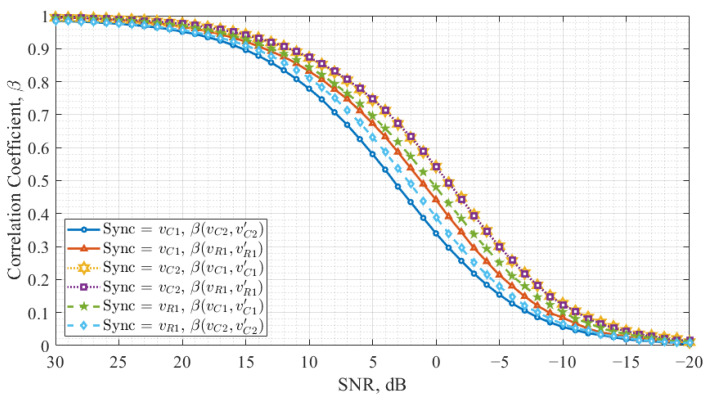
Vilnius chaos oscillator synchronization noise immunity.

**Figure 7 entropy-27-00296-f007:**
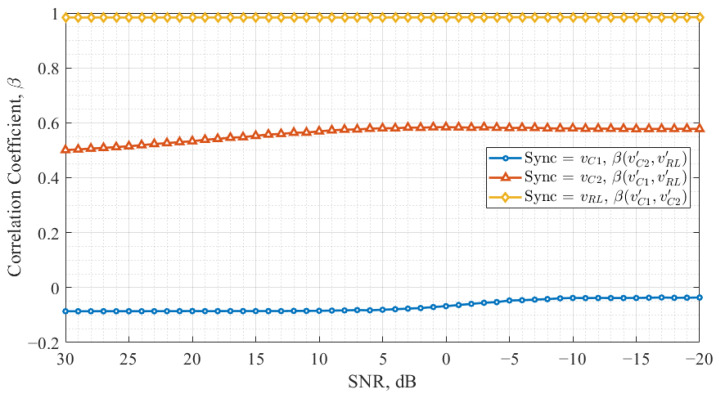
Results of the Colpitts chaos oscillator, showing the signal cross-correlation coefficients under different synchronization configurations.

**Figure 8 entropy-27-00296-f008:**
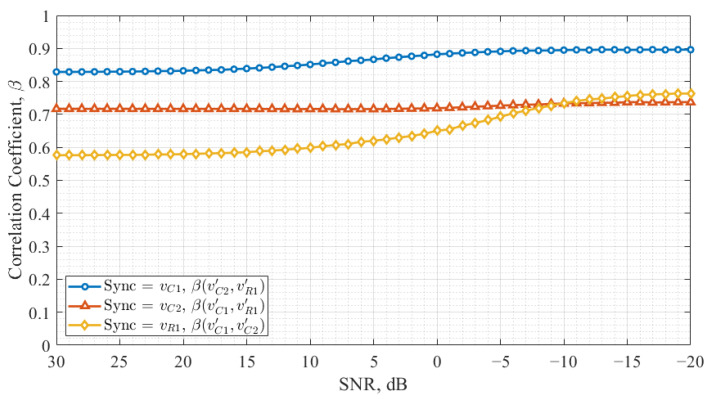
Results of the Vilnius chaos oscillator, showing the signal cross-correlation coefficients for different synchronization signals.

**Figure 9 entropy-27-00296-f009:**
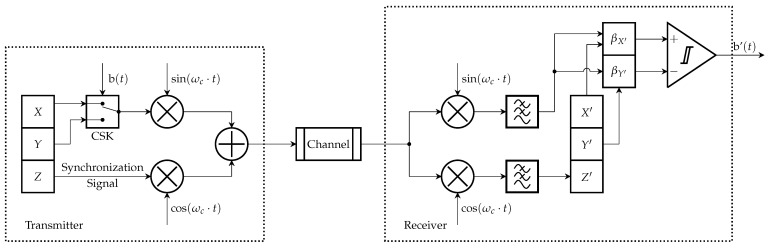
Quadrature chaos shift keying data transmission system block scheme.

**Figure 10 entropy-27-00296-f010:**
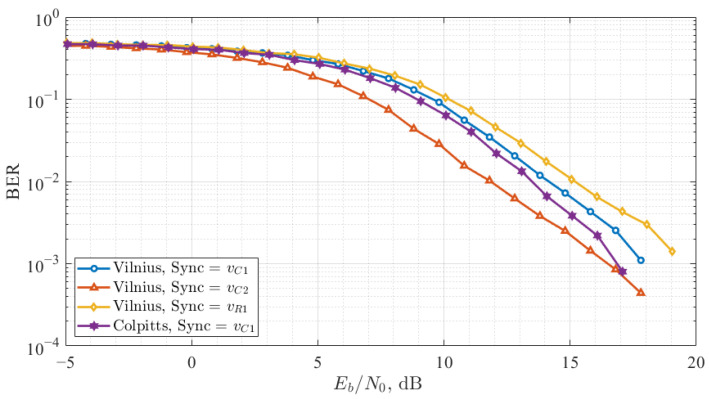
BER versus $E_{b}/N_{0}$ curves of the QCSK data transmission system.

**Table 1 entropy-27-00296-t001:** Colpitts and Vilnius chaos oscillators configurations.

Colpitts Chaos Oscillator			Vilnius Chaos Oscillator		
X	Y	Z	X	Y	Z
$v_{C1}$	$v_{C2}$	$v_{RL}$	$v_{C1}$	$v_{C2}$	$v_{R1}$
$v_{RL}$	$v_{C1}$	$v_{C2}$	$v_{R1}$	$v_{C1}$	$v_{C2}$
$v_{C2}$	$v_{RL}$	$v_{C1}$	$v_{C2}$	$v_{R1}$	$v_{C1}$

**Table 2 entropy-27-00296-t002:** Error count for the QCSK configuration without and with the detection threshold in a system based on the Vilnius chaos oscillator using $v_{R1}$ as the synchronization signal.

SNR, dB	Without Threshold			With Threshold		
	BER	False ‘0’ Count	False ‘1’ Count	BER	False ‘0’ Count	False ‘1’ Count
−20	0.4835	2414	2421	0.4810	2416	2394
−15	0.4407	2287	2120	0.4370	2168	2202
−10	0.3249	1761	1488	0.3195	1598	1597
−5	0.1134	702	432	0.1053	519	534
0	0.0121	79	42	0.0107	57	50
2	0.0048	34	14	0.0043	21	22
4	0.0019	12	7	0.0014	7	7

## Data Availability

Data is contained within the article.

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
