# Peer review of "Design Particularities of Quadrature Chaos Shift Keying Communication System with Enhanced Noise Immunity for IoT Applications"

_entropy, 2025, doi:10.3390/e27030296_

Round 1
Reviewer 1 Report
Comments and Suggestions for Authors
This paper investigated the synchronization noise immunity of Vilnius and Colpitts chaos oscillators with different configurations. Results show that the Vilnius chaos oscillator has more configurations than the Colpitts chaos oscillator to support high-quality synchronization. Based on the synchronization configurations, a QCSK communication system was implemented. This paper is well written, and the presented results are convincing. Publication is recommended after addressing the following issues.
- Compared to Vilnius chaos oscillator, only one out of three configurations supports the synchronization with a high correlation coefficient for Colpitts chaos oscillator. It is interesting to know why this happens.
- For both Colpitts chaos oscillator and Vilnius chaos oscillator, there exist configurations which show a slight increase of correlation coefficient as the decrease of SNR. Comments should be made to explain this.
- As shown in Fig. 10, BER of the QCSK data transmission system decreases as a function of Eb/N0. A threshold of BER favoring the transmission should be given.
Author Response
This paper investigated the synchronization noise immunity of Vilnius and Colpitts chaos oscillators with different configurations. Results show that the Vilnius chaos oscillator has more configurations than the Colpitts chaos oscillator to support high-quality synchronization. Based on the synchronization configurations, a QCSK communication system was implemented. This paper is well written, and the presented results are convincing. Publication is recommended after addressing the following issues.
- Compared to Vilnius chaos oscillator, only one out of three configurations supports the synchronization with a high correlation coefficient for Colpitts chaos oscillator. It is interesting to know why this happens.
Thank you for your insightful comment. The difference in synchronization performance between the Colpitts and Vilnius chaos oscillators can be attributed to their fundamental circuit topologies, nonlinear elements, and state variable interactions. While both oscillators exhibit chaotic behavior, their dynamical properties and sensitivity to initial conditions vary significantly.
The Colpitts oscillator is fundamentally a feedback-based oscillator with a capacitive voltage divider influencing the nonlinear behavior. In the synchronization process, the choice of synchronization node impacts how signal propagates through the circuit. Since the oscillator’s state variables are tightly coupled, some configurations may introduce stronger mismatches in the response system, preventing effective synchronization.
In contrast, the Vilnius chaos oscillator has a more distributed nonlinear dynamic structure, where each state variable contributes more independently to the system’s overall behavior. This structure allows multiple synchronization paths, leading to a higher probability of achieving stable synchronization across different configurations.
Thus, results suggest that the Colpitts oscillator exhibits a more restrictive synchronization behavior due to its circuit topology, nonlinear characteristics, and strong variable coupling, which limit the effectiveness of synchronization in some configurations. In contrast, the Vilnius chaos oscillator provides more flexible synchronization possibilities, allowing multiple high-correlation configurations.
We appreciate this valuable comment, as it helped us clarify this aspect of our study. Based on this feedback, we have made modifications to the manuscript and included a shorter version of this explanation in Section 3.2 to provide better clarity on the observed synchronization differences.
- For both Colpitts chaos oscillator and Vilnius chaos oscillator, there exist configurations which show a slight increase of correlation coefficient as the decrease of SNR. Comments should be made to explain this.
We appreciate this insightful observation. The observation that the correlation coefficient slightly increases as the SNR decreases in certain configurations for both Colpitts and Vilnius chaos oscillators can be attributed to two main factors: the impact of noise on the chaotic synchronization process and the nature of correlation measurement in low-SNR conditions.
In some chaotic systems, a phenomenon known as stochastic resonance (SR) or noise-enhanced synchronization can occur, where a moderate level of noise helps stabilize synchronization by forcing the response system to align with the drive oscillator more consistently. (https://www.researchgate.net/publication/262452743_Noise-Enhanced_Synchronization_of_Stochastic_Magnetic_Oscillators https://journals.aps.org/pre/abstract/10.1103/PhysRevE.58.7118)
In the Colpitts and Vilnius chaos oscillators, noise may be a perturbation that prevents transient divergence in the response system. This effect can slightly increase the correlation coefficient under specific SNR conditions before synchronization deteriorates at very low SNR.
The configurations exhibiting this effect may correspond to cases where one of the state variables has a stronger dependence on external perturbations, making it more responsive to noise-induced stabilization.
The slight increase in the correlation coefficient at lower SNR levels is likely caused by a combination of noise-induced stabilization. While noise generally degrades synchronization, in some configurations, it can temporarily enhance alignment between drive and response signals, leading to a short-term increase in measured correlation values before full desynchronization occurs at very low SNRs.
- As shown in Fig. 10, BER of the QCSK data transmission system decreases as a function of Eb/N0. A threshold of BER favoring the transmission should be given.
Thank you for your insightful comment. In wireless communication and IoT applications, acceptable Bit Error Rate (BER) thresholds vary depending on specific system requirements and the effectiveness of error correction codes (ECC). Generally, a BER below 10−3 is considered sufficient, as ECC can effectively detect and correct errors at this level, ensuring reliable data transmission. This is why we evaluate BER up to this threshold, as it represents the point where ECC can be applied to enhance system performance.
Reviewer 2 Report
Comments and Suggestions for Authors
The title seems to be too long and shift the focus away.
A good review on IoT security was given and led to the PLS and DCSK discussions. A good discussion to the two oscillators are given and provide a very good insight. Some good recommendations are made based on good result analysis.
However, the discussion to the proposed QCSK system seems to be very brief or shorter, comparing to the previous discussions. The design and testing date of the mentioned experiments are not clear. It will be good to explain how experiments are designed and conducted, as well as how results were collected.
It will also be helpful to explain how can the CSK, DCSK and QCSK be used for real world applications or scenarios.
Author Response
- The title seems to be too long and shift the focus away.
We truly appreciate your suggestion regarding the title length and its potential impact on the paper's focus. However, we have chosen the current title to fully represent the system under research, ensuring that the full name of the QCSK communication system is explicitly stated. Additionally, we aimed to highlight the study's focus on improving noise resilience in chaos-based data transmission systems while emphasizing the design particularities explored in our research.
- A good review on IoT security was given and led to the PLS and DCSK discussions. A good discussion to the two oscillators are given and provide a very good insight. Some good recommendations are made based on good result analysis.
Thank you for your positive feedback. We appreciate your recognition of the IoT security review, the discussion on PLS and DCSK, and the insights provided on the chaos oscillators. We are also grateful that you found the result analysis and recommendations valuable. Your comments reinforce the significance of our study, and we are pleased that our findings contribute meaningfully to the field.
- However, the discussion to the proposed QCSK system seems to be very brief or shorter, comparing to the previous discussions. The design and testing date of the mentioned experiments are not clear. It will be good to explain how experiments are designed and conducted, as well as how results were collected.
We appreciate your observation regarding the briefness of the QCSK system discussion compared to previous sections. To enhance clarity, we have expanded the QCSK system design and experimental methodology description to provide a more detailed explanation of how the system was implemented and evaluated.
To address this, we have added further details to the introduction, ensuring that the methodology and data collection process are clearly outlined. We believe these refinements improve the manuscript's comprehensiveness while maintaining its focus.
- It will also be helpful to explain how can the CSK, DCSK and QCSK be used for real world applications or scenarios.
Addressing your valuable comment, we clarified the real-world applicability of chaos-based communication schemes in the article. We have added a brief clarification in the manuscript's introduction to address this.
Chaos-based communication systems, such as CSK, DCSK, and QCSK, present a promising solution for data transmission in sensor networks, particularly in low-power IoT applications. Since sensors often have limited processing capacity and power constraints, chaos-based modulation techniques offer an advantage due to their simple circuit implementation. These properties make them well-suited for transmitting data from distributed sensors to larger system nodes.
We believe this addition provides a clearer connection between chaotic modulation techniques and their potential real-world applications while maintaining the focus of the paper.
Round 2
Reviewer 1 Report
Comments and Suggestions for Authors
All my concerns have been addressed appropriately, publication is recommended.